# Mutually Beneficial Combination of Molecular Dynamics Computer Simulations and Scattering Experiments

**DOI:** 10.3390/membranes11070507

**Published:** 2021-07-05

**Authors:** Nebojša Zec, Gaetano Mangiapia, Alex C. Hendry, Robert Barker, Alexandros Koutsioubas, Henrich Frielinghaus, Mario Campana, José Luis Ortega-Roldan, Sebastian Busch, Jean-François Moulin

**Affiliations:** 1German Engineering Materials Science Centre (GEMS) at Heinz Maier-Leibnitz Zentrum (MLZ), Helmholtz-Zentrum Hereon, Lichtenbergstr. 1, 85748 Garching bei München, Germany; nebojsa.zec@hereon.de (N.Z.); gaetano.mangiapia@hereon.de (G.M.); 2School of Biosciences, University of Kent, Canterbury CT2 7NJ, UK; ach49@kent.ac.uk (A.C.H.); J.L.Ortega-Roldan@kent.ac.uk (J.L.O.-R.); 3School of Physical Sciences, University of Kent, Canterbury CT2 7NH, UK; R.Barker@kent.ac.uk; 4Jülich Centre for Neutron Science (JCNS) at Heinz Maier-Leibnitz Zentrum (MLZ), Forschungszentrum Jülich, Lichtenbergstr. 1, 85748 Garching bei München, Germany; a.koutsioumpas@fz-juelich.de (A.K.); h.frielinghaus@fz-juelich.de (H.F.); 5ISIS Neutron and Muon Facility, Rutherford Appleton Laboratory, Science & Technology Facilities Council, Didcot OX11 0QX, UK; mario.campana@stfc.ac.uk

**Keywords:** neutron reflectometry, X-ray reflectometry, small-angle neutron scattering, small-angle X-ray scattering, molecular dynamics simulations, scattering length density profile, phospholipid membrane

## Abstract

We showcase the combination of experimental neutron scattering data and molecular dynamics (MD) simulations for exemplary phospholipid membrane systems. Neutron and X-ray reflectometry and small-angle scattering measurements are determined by the scattering length density profile in real space, but it is not usually possible to retrieve this profile unambiguously from the data alone. MD simulations predict these density profiles, but they require experimental control. Both issues can be addressed simultaneously by cross-validating scattering data and MD results. The strengths and weaknesses of each technique are discussed in detail with the aim of optimizing the opportunities provided by this combination.

## 1. Introduction

Phospholipid-based bilayers are the main components of biological membranes and represent their basic structural elements [1]. The main role of the cell membrane is to protect the cell from its surroundings, allowing it to have a well-defined environment and accomplish its vital functions [2]. Given the importance of the membrane, structural details for the cell biology, several characterization methods have been used to investigate the structure under different conditions (microscopy [3,4], spectroscopy [5,6], scattering methods [7,8,9,10] and simulations [11,12,13,14]). Of special interest in this paper are the scattering methods that give access to the structure and dynamics of the system under investigation. These methods are non-invasive, non-destructive over the duration of the data collection and probe a large sample volume, thus providing statistically relevant information [15]. The typical membrane length scales are relatively large compared to atomic dimensions, hence the focus of this work is on scattering by large-scale structures which can be investigated by reflectometry and small-angle scattering methods.

Fundamentally, two main types of probe can be used for these scattering experiments: X-rays and neutrons. Laboratory X-ray sources provide the possibility of performing many useful experiments and high-flux synchrotron sources make very fast and extremely sensitive measurements feasible. Neutron scattering experiments can only be performed at large-scale facilities, but they nevertheless play a fundamental role in the landscape of membrane characterization methods [16,17,18,19]. In contrast to X-rays, neutrons interact in a non-destructive fashion with the material under examination. Due to their weak interaction with matter, neutrons have a large penetration depth for most materials, allowing for an elaborate sample environment. Being scattered by the nuclei, and not by the electrons as in the case of X-rays, neutrons offer the possibility to add isotopic sensitivity to the measurements. As a great opportunity for biological systems, the neutron scattering power of hydrogen and deuterium differs widely and neutrons are thus extremely sensitive to the distribution of hydrogen in the sample. The most obvious strategy for taking advantage of this property is to perform measurements on membranes prepared in light or heavy water, but more complex (and more costly) isotopic substitution methods targeting specific molecular sites can also be utilized.

All scattering methods provide a description in reciprocal space, which can be understood as the Fourier transform of the structure of the sample. The data thus tell about periodicity and spatial correlation in the sample [20,21]. The interpretation of such data is by no means intuitive and the eventual aim of all measurements is to describe the actual position of all atoms or molecules in the sample. In the process of inverting the information contained in scattering data from reciprocal to real space, problems arise (in particular, the phase problem [21]), which usually hinders finding an unequivocal solution. To tackle this issue, independent information must be found in order to put constraints on the inversion problem. Unfortunately, there is no available experimental method offering the needed spatial resolution over the required length scales. Nevertheless, computer simulations in the framework of Molecular Dynamics (MD) provide invaluable insights in the real space structure of these complex systems.

MD simulations applied to phospholipid membranes provide an atomic-level description of the system. The positions of individual atoms are followed by numerically solving classical equations of motion. Therefore, MD simulations provide atomic resolution unavailable to the experiments presented here. Combining MD simulations and scattering experiments is beneficial for studying phospholipid membranes but can also be used for the structural analysis of completely unrelated systems [22].

MD and the experimental methods described here (SAS and reflectometry) probe the sample’s structure over a limited range of length scales. Those ranges overlap, and, hence, cross-validation is only possible over this restricted domain [23]. Another important aspect to consider is the fact that an MD simulation typically only describes a very brief time interval while the integration times used for data acquisition in NR and SANS are orders of magnitude longer (from seconds to hours). Similarly, simulations typically cover some cube nanometers, while experiments tend to average over cube millimeters. Precautions must thus be taken to ensure that MD simulations do not merely describe transient structures which would be averaged out in the measurements. Conversely, simulations give access to the Ångström scale, which is not directly probed by these experimental techniques.

In classical MD simulations, the interaction potential energy is described in the form of a force field, based on both empirical and quantum chemical data. Validation of the force-field parameters is a tedious and challenging task, so online topology and force-field parameter builders have become popular as a simple solution [24,25,26]. One has to be very critical of the parameters obtained in this way and ensure that the theoretical model and applied methodology describe the molecule of interest “reasonably well”. On the other hand, besides lower spatial resolution compared to the MD and the phase problem, small-angle scattering and reflectometry experiments have additional experimental uncertainties related to the sample and instrumentation. It is therefore theoretically possible that inaccurate experimental data match an incorrect MD simulation perfectly.

To successfully combine these techniques, a certain level of understanding of both scattering experiments and computer simulations is essential in order to fully understand the advantages and limitations of both methods and avoid putting too much (or too little) confidence in the results extracted from one of these methods alone. One can use MD simulation trajectories to extract neutron scattering length density profiles, directly calculate the corresponding reflectivity or small-angle scattering pattern and plan an experiment in order to optimize the use of beam-time at large-scale facilities. One can see the effect of changing different parameters such as instrumental resolution, get a hint as to whether the effect can be experimentally observed and plan an experiment in an effective and efficient manner.

In this article, the study of a single bilayer of DMPC (1,2-dimyristoyl-sn-glycero-3-phosphocholine) and multilamellar SoyPC (mainly composed of 1,2-dilinoleoyl-sn-glycero-3-phosphocholine) is used to showcase the joint use and cross-validation of MD simulation and scattering experiments. DMPC is a double-saturated phospholipid composed of two myristoyl chains, used in many biophysical studies [27,28] and as an excipient in pharmaceutical formulations [29]. SoyPC is a mixture of phospholipids found in soy and used as a model bilayer in some studies aimed at investigating the interaction of cell membranes and active ingredients [30,31]. The aim of this work is not so much to discuss the properties of the selected phospholipids as to describe the methodology of combining simulation and experiment and the challenges behind it. Since there are many things that can go wrong in both, it is important to establish the methodology and find sources of potential errors before focusing on more complex systems.

The approach to combine simulations with scattering experiments is not new; it was for example used to study peptide self-organization into switchable films at an air–water interface by Xue et al. [32] and by Vanegas et al. [33] to study the insertion of the dengue virus envelope protein into phospholipid bilayers. These techniques were also applied to investigate the contact angles and adsorption energies of nanoparticles at the air-liquid interface [34]. Back in 2005, Benz et al. [23] developed a protocol for comparing MD simulations with X-ray (XRR) and neutron reflectivity (NR) and showed that neither the united-atom GROMACS nor the CHARMM22/27 force fields could reproduce experimental data. More recently, a method for producing continuous scattering length density (SLD) profiles from MD simulations has been presented for interpreting reflectivity data from phospholipid bilayers [35]. Koutsioubas [36] performed coarse-grained MD simulations with the standard MARTINI force field and obtained quantitative and semi-quantitative agreement with neutron reflectivity data for DPPC membranes in the liquid and gel phase, respectively. On the other hand, McCluskey et al. [37] observed that the MARTINI potential model did not accurately describe the 1,2-distearoyl-sn-phosphatidylcholine (DSPC) monolayer, while the Berger and Slipid potential models showed better agreement.

Several computer programs for reading MD simulation trajectories, calculating the scattering length density profile and neutron reflectivity, and making direct comparison with the experiment have been developed over the years. SIMtoEXP [38] and NeutronRefTools (as a VMD plug-in) [39] were developed particularly for phospholipid membrane research. The high number of citations shows that these solutions have been accepted and regularly used by the scientific community. Being completely aware of their existence, we employ here a self-written software solution that will be published soon.

In the following, we describe the different techniques, show two examples of phospholipid molecules in two different morphologies and discuss the robustness of experimental features and their constraints on real space structure.

## 2. Background

In order to provide the tools needed in the discussion, this section introduces some fundamentals of scattering theory and puts them in the context of the present problem. Keeping in mind the typical expectations of the computer simulation community, the strong points as well as the pitfalls of the scattering methods are stressed along the way. Momentum transfer $\vec{Q}$, the natural variable against which scattering intensity is measured in an actual experiment, is introduced first. This variable takes the radiation characteristics (wavelength) and the geometrical details of the experiment into account. The SLD, which describes how strongly a given medium will scatter as a function of its composition, is then introduced and used to express the index of refraction, which in turn is used to predict the propagation of neutrons or photons (both considered as waves) in matter and eventually analyze reflectometry and small-angle experiments.

Neutron and X-ray scattering experiments measure the number of scattered neutrons/photons as a function of the vector $\vec{Q}$, which describes the momentum transfer the wave undergoes upon scattering. $\vec{Q}$ is a function of the experiment geometry, which we symbolically represent here by θ, and of the wavelength of the radiation used, λ:(1)$$\vec{Q}\left(\theta ,\lambda \right)=\vec{k_{f}}-\vec{k_{i}}$$
where $\vec{k_{i}}$ and $\vec{k_{f}}$ are the wave vectors of the incident and scattered radiation, respectively.
(2)$$\left|\vec{Q}\right|=Q\propto \frac{2\pi }{l}$$
signifies that the modulus of each *Q* vector in reciprocal space is associated with an inter-distance *l* in direct (real) space, which is characteristic of the size of the scattering structure in the corresponding direction.

The practical problem is to compute the real space sample structure which is compatible with the scattering intensity distribution, measured in the reciprocal or *Q* space. In tackling this task, which is central to the whole crystallography field, the most fundamental obstacle is the phase problem. What the detectors actually measure is the intensity, i.e., square of the amplitude of the scattered waves. Consequently, all information relative to the phase of those waves is irremediably lost. From the measurement, it is therefore impossible to unequivocally deduce the positions of the scattering particles in an absolute way and a given experimental dataset can correspond to a multitude of real-space structures.

There are several ways to work around this ambiguity: First, one can gain additional experimental data by performing measurements for specifically adjusted scattering contrast of the different constituents without affecting the sample’s structure [40]. How this is practically achieved is discussed in detail in the next sections. While this reduces the number of possible real-space structures considerably, a usually unachievable n(n+1)/2 contrasts would have to be measured to be able to solve the real-space structure of *n* components from the data analytically—and even if that many measurements can be performed, experimental imperfections and limited counting statistics limit their usefulness [41].

A second method is to form periodic structures in the system. In the case of membranes, this can for example be achieved by using stacks of bilayers (multilayers) that are periodic in the direction of the membrane normal. This leads to the formation of *Bragg peaks* in the scattered intensity at values of *Q* where the phase is either 0${}^{\circ }$ or 180${}^{\circ }$. In a traditional approach, one would then use only the scattered intensity at the Bragg peak positions where the phase can be determined [42]. Although this approach leaves the whole information contained in the rest of the scattering pattern unused, the SLD profile can be reconstructed precisely if many Bragg peaks are measured. In reality, however, it is only possible to measure ∼2–5 Bragg peaks due to the disorder inherent in the system and experimental limitations, severely limiting the precision of the extracted information. It is also possible to take the whole scattering pattern into consideration (see [43] and references therein), but the presence of Bragg peaks makes the precise measurement of the specular reflectivity between the Bragg peaks somewhat unreliable, as discussed below.

Complementary to these experimental approaches, one can use a theoretical approach in which a real-space model of the system is built with as many external constraints as possible. The MD method is here the instrument of choice. As shown elsewhere, although one needs to take care of several practical details [22,35,38,39], it is relatively straightforward to compute the scattering pattern corresponding to an MD simulated structure and compare it to the experimental data. This goes beyond the normal fitting procedure in which a set of parameters describing the structure is optimized in order to reproduce the data. While this method cannot prove the accuracy of a given model, it can, however, falsify many models, which must not be underestimated.

### 2.1. Scattering Length Density

In the following, we briefly describe the scattering processes and introduce the fundamental concept of scattering length density and its influence on the transmission and refraction of the waves in a medium. At the end of this section, the practical possibility of taking advantage of probe type and isotopic composition to control scattering contrast is apparent.

The treatment for X-rays and neutrons is very similar and differs only in the interaction of the corresponding radiation with matter. Neutrons interact with the nuclei (we leave aside the magnetic interactions with electrons, which is generally less relevant for the study of biological materials) while X-rays, being an electromagnetic wave, interact with the electron cloud. We thus here use the general wording “wave” and show probe-specific expressions only where relevant.

The interaction between a wave and a medium is described in quantum mechanical terms by the average potential $\bar{V}$
(3)$$\bar{V}=\frac{2\pi \hbar^{2}}{m}\rho$$
where *m* is the neutron mass, *ℏ* the Planck constant divided by 2π, and
(4)$$\rho =\frac{1}{\mathrm{volume}}\sum_{j}b_{j}$$
is the so-called scattering length density of the medium (which we also denote by SLD) and is the result of the superposition of all contributions $b_{j}$ (scattering lengths) describing the interaction strength of each individual scatterer *j*.

A general solution of the Schrödinger equation which satisfies the potential $\bar{V}$ and describes the propagation of a wave at every point $\vec{r}$ in the medium is
(5)$$\Psi \left(\vec{r}\right)=A\exp\left(in\vec{k_{0}}\cdot \vec{r}\right)$$
where *n* is the complex index of refraction relating $k_{0}$, the wave momentum in vacuum, and *k*, the momentum it would have in a material medium,
(6)$$n=\frac{k}{k_{0}}\;.$$

The real part of *n* describes the wave phase velocity in the medium, while the imaginary part describes the absorption phenomena by damping the wave intensity, which is the square of the modulus of Ψ. In the neutron case, the absorption is usually negligibly small and *n* is a real number.

Similar to what we experience in everyday life while looking at things, scattering methods give us the ability to distinguish different parts of the samples from each other only if their indices of refraction differ, irrespective of their chemical nature.

For X-rays, the scattering length is proportional to the product of the atomic number *Z* and the classical electron radius or Thomson scattering length $r_{0}\approx 2.82$ fm. The energy dependence of the scattering length, which varies abruptly around absorption edges, is described by semi-empirical atomic scattering factors $f_{1}$ and $f_{2}$, leading to the following expression for the refraction index where *N* denotes the number concentration of the given atom:(7)$$n_{X}=1-\frac{1}{2\pi }Nr_{0}\lambda^{2}\left(f_{1}+if_{2}\right)$$

For neutrons, the energy-independent nuclear scattering length *b* substitutes the electron radius in the previous expression and one gets
(8)$$n_{n^{0}}=1-\frac{1}{2\pi }\lambda^{2}\rho \;.$$

As can be intuitively expected from the nature of their interaction, in the case of neutrons the scattering lengths of different isotopes of the same element differ from each other. This isotopic dependency of *b* is seemingly random [44], but it is very interesting to observe that, in the case of hydrogen and deuterium, the difference is very large, $b_{H}$ being −3.74 fm for hydrogen and $b_{D}$ = 6.67 fm for deuterium.

From those observations, it is clear that X-rays and neutrons will experience a different index of refraction between the components of the sample, thereby introducing a contrast between those regions. As hinted in the Introduction, one can thus obtain additional independent information about the system under investigation by: (a) combining X-ray and neutron measurements; and/or (b) varying the isotopic composition of the sample used for neutron scattering while keeping its chemical composition and structural details essentially unchanged. In the context of molecular biology, it is clear that advantage can be taken of this method by tuning the isotopic composition of the ubiquitous water molecules. By simply mixing H${}_{2}$O and D${}_{2}$O, one can adjust the contrast with precision [45,46]. More complex isotopic substitution schemes, for instance at specific molecular sites, can also be used to achieve more targeted control [47,48].

### 2.2. Reflectometry

Reflectometry takes advantage of the variation of the index of refraction across planar interfaces in order to investigate structural and compositional profiles.

When a wave impinges on a flat and smooth horizontal surface separating two media (denoted 1 and 2), it can be reflected back into the original medium (reflection into 1) or refracted into medium 2. Since the ideal interface we describe is an SLD fluctuation along the vertical direction only, it cannot affect the in-plane components of the incident wave’s momentum. The reflection is purely specular and happens under the same angle as the angle of incidence.

The convenient variable to describe this problem is again the momentum transfer vector $\vec{Q}$, which is here strictly vertical:(9)$$\vec{Q}=\vec{k_{f}}-\vec{k_{i}}=\left|\vec{Q}\right|\cdot \vec{n_{z}}=Q_{z}\cdot \vec{n_{z}}\;,$$
where $\vec{n_{z}}$ is the unit vector along the vertical direction, *z*.

Since we are only considering elastic scattering, the norm of the momentum of the wave is conserved and
(10)$$Q_{z}=\frac{4\pi \sin(\theta )}{\lambda }$$
where θ is the angle of incidence on the surface.

Regarding the refracted wave, since the index of refraction differs in the two media
(11)$$k_{1}/k_{2}=n_{2}/n_{1},$$
and, following the above argument that the in-plane components of *k* cannot change, we get
(12)$$n_{1}\cos\left(\theta_{1}\right)=n_{2}\cos\left(\theta_{2}\right)$$
where $\theta_{1}$ is the angle of incidence and $\theta_{2}$ is the refraction angle, both measured between the surface and the corresponding propagation vector. The above relationship is no other than Snell’s law of optics, which also holds for neutrons and X-rays.

From this relation, if the index of refraction is smaller than 1 (often the case for neutrons and X-rays), there exists a critical *Q* below which $\theta_{2}$ will be zero, i.e., below which the incident wave will undergo total reflection by the interface:(13)$$Q_{c}=4\sqrt{\pi (\rho_{1}-\rho_{0})}$$

The amplitude reflectivity (*r*) and amplitude transmitivity (*t*) of the surface are given by the Fresnel relationships, which can be derived from continuity conditions. Applying the small-angle approximation, which holds in the case of reflectometry measurements, leads to [49]
(14)$$\begin{array}{cc} r & =\frac{A_{\mathrm{reflected}}}{A_{\mathrm{incident}}}=\frac{\theta_{\mathrm{incident}}-\theta_{\mathrm{refracted}}}{\theta_{\mathrm{incident}}+\theta_{\mathrm{refracted}}}\;\mathrm{and} \end{array}$$
(15)$$\begin{array}{cc} t & =\frac{A_{\mathrm{refracted}}}{A_{\mathrm{incident}}}=\frac{2\theta_{\mathrm{incident}}}{\theta_{\mathrm{incident}}+\theta_{\mathrm{refracted}}}\;, \end{array}$$
where *A* represents the respective amplitudes.

In the case of stratified media on a semi-infinitely thick substrate, a valid description of practical experiments one might perform to study supported thin films, the impinging wave can undergo reflection or refraction at each interface. The wave emerging from the surface is the superposition of all the waves which have traveled paths through the sample that do not end up being transmitted into the semi-infinite substrate.

Similar to the simple case of a single interface, one can express the amplitude reflectivity and the amplitude transmitivity via the Fresnel equations. Starting from the semi-infinite substrate where no multiple reflections are to be considered, one can then recursively reconstruct the reflectivity at the topmost surface. This method, which leads to an exact result, was introduced by Parratt [50]. A computationally convenient method based on the formalism of optical transfer matrices was independently proposed by Abelès [51,52] and leads to the same exact result.

If the interface between regions of different SLD is diffuse rather than sharp as assumed above, two approaches can be used for the evaluation of the reflectivity. The first approximation was proposed by Névot and Croce [53]. It introduces an interfacial roughness factor which damps the reflected waves and which is expressed in a similar way as the Debye–Waller factor describing, in crystallography, the effect of the thermal motion blurring the atomic positions and thereby lowering the diffracted intensities. In this model, the position of the interface is described as normally distributed around its nominal position with a given standard deviation σ. The corresponding SLD profile is a smooth transition from one SLD to the next in a sigmoidal step described by the error function associated to the standard deviation. This approach has the obvious advantage of describing the diffuse interface by a single number. However, it has to be stressed that this approximation is only valid if the roughness is much smaller than the layer thickness.

The second method used to deal with diffuse interfaces, while more computationally demanding, makes it possible to describe arbitrary SLD profiles. In this second approach, the SLD profile is simply discretized into bins thin enough to ensure that they can be considered to be of constant SLD. The reflectivity computation can then follow without further approximation by means of either the Paratt or the Abeles algorithm. Although this approach is potentially able to better “follow” the actual SLD profile and can deal with diffuse areas too broad to be safely described by a Nevot–Croce roughness parameter, it lacks the ability to condense structural information in simple and clear parameters such as layer thickness, width of a transition region, etc. Such a convenient description can of course nevertheless be obtained *a posteriori* by adjusting an analytical model to the binned SLD profile used for the simulations.

It should be kept in mind that, whatever the chosen approach used to describe diffuse SLD transition regions, different lateral distributions of matter could lead to the same SLD profile along the vertical. The in-plane fluctuations, which have been averaged out here, would be the cause of the off-specular scattering. The two types of reflections—specular and off-specular—can be easily understood with the help of an everyday-life analogy: the specular (literally, mirror-like) behavior is what we observe when we contemplate the sunset reflection on the surface of a perfectly still lake, on a windless evening. This image of an undeformed sun tells us that the surface of the water is perfectly flat (and reflective). If we repeat this contemplation while a strong wind is blowing, we will only be able to see a blurry image of the sun on the water surface: the lateral structures on the surface, its roughness, will reflect the light away from the expected ideal trajectory, hence in an off-specular or non-specular way. A detailed analysis of the blurred image could lead to an understanding of the details of the wavy surface. Practical implications of the presence of off-specular scattering are discussed briefly when dealing with actual measurements.

The description of the specular reflectivity evaluation for a given SLD profile given above is exact and can be used for numerical evaluations. It is, however, interesting to keep the results obtained in the framework of the first Born approximation in mind, i.e., in the limit of weak scattering. In this case, one gets the “master-equation of reflectivity” [49]:(16)$$R=R_{\mathrm{Fresnel}}{\left|\int \frac{d\rho }{dz}\exp\left(iQz\right)dz\right|}^{2}$$
which relates the reflectivity of an interface with arbitrary interfaces to the reflectivity of a multilayer with sharp interfaces ($R_{\mathrm{Fresnel}}$) and the spatial rate of change of the SLD, ρ.

The Born approximation clearly does not hold for small values of *Q*, for which many reflections or even total reflection take place, but it can be used to gain intuition about the reflectivity observed at large *Q*. This expression of scattering as a Fourier transform of real space makes it clear that the reflectivity curve of a layer of thickness *l* will display oscillations as a function of *Q* having a period given by 2π/l. In the case of periodic structures such as those encountered, for instance, in multilayered phospholipids, intensity will build up at specific locations in *Q* space and appear as the Bragg peaks known in diffraction. Moreover, it is clear from this expression that only regions which display an SLD contrast (i.e., where the derivative of the SLD is not zero) will contribute to the reflectivity. Last but not least, this Fourier-transform approach also helps understand the origins of the spatial resolution limits of the scattering methods: the maximal observed *Q* value will determine the size of the smallest object which can be resolved by a scattering experiment.

Reflectometry is the method of choice when focusing on planar surfaces or buried interfaces. The sample consists of ∼10 cm${}^{2}$ substrate covered with a sample layer, resulting in very low amounts of sample required for an experiment. The measurement geometry means that one is exclusively sensitive to the direction along the interface normal in specular scattering and can get separate information about in-plane correlations through off-specular scattering.

### 2.3. Small-Angle Scattering

Differently from reflectometry, which probes the characteristics of planar interfaces, in a small-angle scattering (SAS) experiment, the characteristics of scattering objects (gels, polymer blends, porous structures, micelle aggregates, etc.) are measured in bulk [54]. In SAS geometry, a collimated beam hits a sample, such as an aqueous solution or a solid, and is (elastically) scattered. As the name indicates, only scattering at low angles (≤30 deg) is recorded by a detector. For isotropic samples, the scattering pattern has no azimuthal dependence and depends uniquely on the modulus of the vector $\vec{Q}$, Q=4πsin(2θ/2)/λ, where 2θ is the scattering angle. From reduction of the experimental data, an important quantity, namely the scattering cross section dΣ/dΩ, is obtained as a function of $\vec{Q}$. This quantity represents the ratio between the number of particles (photons or neutrons) that in the unit of time are scattered in a certain direction reaching a solid angle element dΩ and the product between the flux of the incident particles on the sample and the value of the solid angle element itself. dΣ/dΩ provides important information about the shape of the scattering structures inside the sample, as well as on the inter-particle interactions [55].

In contrast to reflectometry, the first-order Born approximation is used for the evaluation of SAS data over the whole *Q* range since multiple scattering effects can usually be neglected. This simplifies data evaluation since, under this approximation, dΣ/dΩ may be expressed as the square modulus of the Fourier transform of the SLD profile $\rho \left(\vec{r}\right)$ [56]:(17)$$\frac{d\Sigma }{d\Omega }={\left|\rho \left(\vec{r}\right)\exp\left(i\vec{Q}\vec{r}\right)d^{3}\vec{r}\right|}^{2}$$

For the case of scattering from objects with spherical symmetry, integration may be carried out in spherical coordinates and Equation (Equation 17) may be simplified to:(18)$$\frac{d\Sigma }{d\Omega }={\left|4\pi \int \rho \left(r\right)r^{2}\frac{\sin(Qr)}{Qr}dr\right|}^{2}$$
which can be used to simulate the cross section starting from the knowledge of the SLD profile obtained from MD.

Compared to reflectometry, where the surface to be probed is suitably prepared on an optically smooth surface, SAS experiments are performed in bulk. The sample is therefore certainly not perturbed by the addition of a substrate. The absence of a substrate also means that it does not have to be described in the model to evaluate the data. Last but not least, the sample preparation is typically easier than the preparation of samples used in reflectometry, where many experimental efforts must be provided to deposit a layer on the substrate.

### 2.4. Molecular Dynamics Simulations

There are several ways to perform atomistic or coarse-grained computer simulations of phospholipid membranes, in particular using Monte-Carlo (MC) or Molecular Dynamics (MD) approaches. In both cases, the interaction potentials between all atoms in the system have to be defined in a force field. There are two parts to a force field: the functional forms of the potentials (e.g., exponential or polynomial) and the parameters in these functions. The choice of the appropriate force field (all-atom, united-atom or coarse grained) and its parameters is the crucial step in every MD simulation. Among many available force fields (AMBER, GROMOS, OPLS, CHARMM, etc.) and their variations, the one validated against the reliable experimental data for the molecules of interest has to be used [57]. If there is no reliable force-field validation data in the literature or if the simulation does not reproduce experimental data, non-trivial force-field parameterization is required. For generating multi-component lipid membrane configurations for MD simulations, there are the MemGen web server [58] and Packmol package [59].

The simulations necessarily simplify the system enormously; a striking example is the contraction of the atoms’ electron clouds into usually fixed point-like partial charges, hereby removing, *inter alia*, polarizability effects. The simulations can therefore not be expected to reproduce all the properties of the membrane at the same time. The art of creating a force field is therefore to tune the functions and parameters such that the quantities of interest are reproduced while others can be incorrect.

MD simulations produce trajectories depicting the motions of atoms over a specified simulation time, usually on the nanosecond to microsecond timescale—depending on the force-field complexity and available computational resources. Some of the most important analyses, technical challenges and existing protocols that can be performed on MD trajectories of the phospholipid membrane were reviewed by Moradi et al. [60]. However, biological processes related to phospholipid membranes are complex and usually challenging either from an experimental or computational aspect. This comprises membrane pore formation, membrane fusion, stalks, domains and curvatures [11,12,61].

## 3. Materials and Methods

### 3.1. Materials

For the experiments on a single supported bilayer of DMPC (1,2-dimyristoyl-sn-glycero-3-phosphocholine, Avanti polar lipids), lipids were dissolved in chloroform followed by solvent evaporation under a stream of nitrogen gas. The lipids were subsequently dissolved in 50 mM HEPES, 50 mM NaCl pH 7.3 buffer followed by sonication to produce vesicles, before being pumped across the reflectivity cell to form a continuous bilayer in 50 mM HEPES, 50 mM NaCl pH 7.3 buffer.

The substrate for the DMPC bilayer consisted of a highly polished silicon block coated with a natural silicon oxide layer. The reflectivity cell was connected to a system where a HPLC pump was used to run the buffers through the sample. Four buffer contrasts were used in these experiments: H${}_{2}$O with $\mathrm{SLD}=-0.56\times 10^{-6}$ Å${}^{-2}$, D${}_{2}$O ($\mathrm{SLD}=6.35\times 10^{-6}$ Å${}^{-2}$), silicon matched water (SiMW) composed of 38% D${}_{2}$O and 62% H${}_{2}$O ($\mathrm{SLD}=2.07\times 10^{-6}$ Å${}^{-2}$) and 4-matched water (4 MW) composed of 66% D${}_{2}$O and 34% H${}_{2}$O, ($\mathrm{SLD}=4.00\times 10^{-6}$ Å${}^{-2}$). The mass of DMPC in the neutron beam during the reflectometry experiment was on the order of 1 μg.

The experimental procedures used to prepare multilayers of SoyPC have been discussed elsewhere [31] together with the chemicals used. Briefly, the phospholipid mixture was dissolved in pure isopropanol and the resulting solution was poured on top of an ultra-polished silicon mirror. The solvent was then removed by keeping the mirror at first at reduced pressure and then under vacuum for a few hours. The mirror was then mounted into a custom-made sample cell and filled with heavy water. In order to visually inspect the SoyPC layer and check for eventual air bubbles formed after injection of D${}_{2}$O, the cell was equipped with a glass cover. Samples used for SANS investigations were prepared starting from a stock solution, dissolving a suitable amount of SoyPC in pure chloroform. The dissolution was favored by a slight warming (40 ${}^{\circ }$C) and a very short sonication treatment (≈5 min). A thin film was subsequently obtained through slow evaporation of the chloroform in a stream of argon, in order to prevent phospholipid oxidation. The phospholipid film was hydrated with D${}_{2}$O, and the resulting suspension was vortexed and then gently sonicated (≈30 min). An aliquot was then repeatedly extruded through a polycarbonate membrane of 100 nm pore size 11 times. The concentration of the hydrogenated SoyPC in D${}_{2}$O was 5.0 mmol/kg. The mass of SoyPC in the neutron beam during the SANS experiment was on the order of 1 mg. During the SANS experiment, the sample was contained in a closed Hellma 404-QX quartz cell that had a thickness of 2 mm, to prevent solvent evaporation.

### 3.2. Reflectometry

The neutron reflectivity measurements on DMPC were taken at the ISIS and Muon Source at the Rutherford Appleton Laboratory, Harwell Science and Innovation Centre, using the time-of-flight SURF instrument [62]. The neutron wavelength ranges from 0.5 to 7 Å, a *Q* range between ∼0.01 and 0.3 Å ${}^{-1}$ was obtained by measuring three different angles $\theta_{\mathrm{incident}}=0.35^{\circ }$, 0.65${}^{\circ }$ and 1.5${}^{\circ }$. The slits were chosen to ensure a footprint of 30 mm by 60 mm at the sample stage with an angular resolution of dQ/Q = 3.5%. Vertical slits were scaled linearly with angle. The time-of-flight spectra were recorded with a ${}^{3}$He point detector [63].

Specular and off-specular reflectivities of the SoyPC multilayer were measured at the vertical reflectometer MARIA [64,65] at Heinz Maier-Leibnitz Zentrum (MLZ) in Garching, Germany, as detailed elsewhere [31]. A neutron beam with an average wavelength λ = 10.0 Å and a wavelength spread of Δλ/λ = 0.10 was used. A 4.1 m collimation length with entrance and exit openings of 1.0 mm was used to collimate the incident beam. The sample was mounted on a goniometer and aligned. Reflectivities were measured by varying the incident angle and recording the pattern of the scattered neutrons with a two-dimensional ${}^{3}\mathrm{He}$ position sensitive detector positioned at 1.9 m from the sample. The experiments were carried out at room temperature.

### 3.3. Small-Angle Scattering

A SANS measurement on SoyPC liposomes was carried out at the KWS-1 diffractometer [66] installed at the Heinz Maier-Leibnitz Zentrum (MLZ) in Garching, Germany. As detailed elsewhere [31], neutrons with average wavelengths of λ = 5.0 Å and a wavelength spread Δλ/λ = 0.10 were used, by means of a mechanical velocity selector. A two-dimensional 128 × 128 array ${}^{6}\mathrm{Li}$ scintillation position sensitive detector measured neutrons scattered from the sample. Three collimation (C)/sample-to-detector (D) distances (namely, C_8_/D_2_, C_8_/D_8_ and C_20_/D_20_, with all distances in meters) allowed collection of data in the scattering vector modulus $Q=4\pi \sin\left(2\theta /2\;\right)/\lambda \;$ ranging between 0.0012 and 0.43 Å^−1^, with 2θ being the scattering angle. The investigated sample was kept under measurement for a period so as to have ≈2 million counts of neutrons. The obtained raw data were corrected for background and empty cell scattering and were then radially averaged. Detector efficiency corrections and transformation to absolute scattering cross sections were executed using a secondary plexiglass standard [67].

### 3.4. Molecular Dynamics Simulations

MD simulations were carried out using GROMACS 2018.1 package [68]. Initial configurations were generated using Packmol [59].

1,2-dilinoleoyl-sn-glycero-3-phosphocholine simulations were carried out in a fully flexible simulation cell containing two phospholipid bilayers consisting of 128 molecules per bilayer (64 molecules per sheet) and 3000 SPC water molecules between the layers was simulated at NPT conditions using Parrinello–Rahman pressure coupling and Nosé–Hoover temperature coupling. The pressure was set to 1 atm through a semi-isotropic coupling with the x/y isothermal compressibility set to $4.5\times 10^{-5}\;{\mathrm{bar}}^{-1}$, while the phospholipids and water were independently coupled to thermal baths at 300 K with a coupling constant of 0.1 ps. The simulations were run for 100 ns with a time step of 1 fs. The equations of motion were integrated using the Verlet leap-frog algorithm. The long-range electrostatic interactions after a cut-off distance at 0.8 nm were accounted for by the particle-mesh Ewald (PME) algorithm [69]. The 12-6 Lennard–Jones interactions were treated by the conventional shifted force technique with a switch region between 1.2 and 1.4 nm. Cross-interactions between different atom types were derived using the standard Lorentz–Berthelot combination rules. United-atom GROMOS 54A7 force-field parameters were used. The model includes 63 atoms (as opposed to to 134 atoms for the all-atom model) since the hydrogen atoms are integrated into the heavy atoms. Periodic boundary conditions (PBC) were applied in all dimensions. The first step of the simulation was an equilibration process for 5 ns. After that, 100 ns of NPT simulation were performed, saving coordinates every 2 ps for analysis.

DMPC (1,2-dimyristoyl-sn-glycero-3-phosphocholine) simulations of 128 DMPC phospholipids and 3655 SPC water molecules were performed with the Berger parameters [70], with the coordinate, force-field and topology files distributed by D. Peter Tieleman (http://wcm.ucalgary.ca/tieleman/downloads, accessed on 1 June 2021). Twenty nanoseconds of NPT simulation were performed with 1 fs time steps, saving coordinates every 2 ps. The resulting area per phospholipid was found to be 60 Å${}^{2}$, which is the same value as the one obtained by Darré et al. [39] with the CHARMM36 force field and TIP3P water model.

Snapshots were rendered in VMD [71]. The trajectories were either analyzed using TRAVIS-1.14.0 [72,73] and Python scripts written in-house or a new Python program dedicated to this purpose, *Made2Reflect*. This approach allows automatizing analysis of very large trajectories (20–100 ns, i.e., ∼20–30 GB in .pdb format), consequently improving statistics and the calculation of scattering length density profiles on the 10–30 min timescale. Using Python makes the script flexible and easy to adjust to the specific needs of monolayers, multilayers, substrates, etc. In Travis, the density profile function (DProf) was used to calculate the number density distribution of particles along the *z* axis, i.e., the direction perpendicular to the phospholipid membrane. The result is a histogram that gives the particle density of a selected particle type (either in nm${}^{-3}$ or relative to uniform density) in thin slices of the system perpendicular to the chosen vector. The distribution is calculated for every molecule and each atom type. In the next step, the number distribution was multiplied by the atomic scattering length obtaining a scattering length density profile. Since the number distribution is calculated for each atom, it allows for a selective deuteration, i.e., selective isotopic substitution, simply by multiplying the number distribution of selected atoms by the scattering length of D instead of H.

## 4. Results

### 4.1. Single Bilayer Neutron Reflectivity of a DMPC Bilayer

To model the scattering of a single DMPC bilayer, a scattering length density profile has to be constructed in real space. This can be achieved either by using an analytic approach where the number densities of different atom types are approximated, e.g., by a Gaussian, or using a numerical approach such as the discretized number density generated from MD simulations. We use here the discretized number density profile of each atom type calculated from the MD simulation so that one can plot and visualize the distribution of the single atom type, specific molecule or its parts. Figure 1 shows the number densities of different elements extracted from an MD simulation, summed up for phospholipid heads and tails separately as an example. Many snapshots along the trajectory were sliced into fine bins (with 0.13 Å thickness) along the *z* axis, the membrane normal, and the different elements/isotopes were histogrammed in these bins. The time average (using the full length of the trajectory) was taken and the number densities of each atom type were multiplied with their respective neutron scattering lengths. The sum of all contributions, i.e., the total scattering length density profile, is also shown in this figure for different H/D substitutions of the water, i.e., contrast variation—H${}_{2}$O, D${}_{2}$O, water with a scattering length density matched to the one of silicon (SiMW, $2.07\times 10^{-6}$ Å${}^{-2}$) and water with a scattering length density matched to be $4\times 10^{-6}$ Å${}^{-2}$ (4 MW).

The first validation step of the calculated SLD profile is to compare the numerical H${}_{2}$O, D${}_{2}$O, SiMW and 4MW SLD values with the theoretical bulk SLD values given as dashed lines on the right-hand side of Figure 1. If these were mismatched, either the density obtained from the simulation or the SLD calculation would be incorrect. The next step is to model a semi-infinite silicon substrate with a native SiO${}_{2}$ layer. The SLD, thickness and roughness of this layer must be obtained through NR measurements and subsequent modeling of Si/SiO${}_{2}$/D${}_{2}$O and Si/SiO${}_{2}$/H${}_{2}$O. The very same characterized silicon wafer is then used for measuring the NR of the phospholipid bilayer. The modeled substrate is given with dashed lines on the left-hand side of the SLD profile (Figure 1). Merging the simulation SLD with the solid substrate SLD has to be performed with caution since one can produce unwanted artefacts in the reflectivity curve [35]. Particular attention has to be given to the treatment of the substrate roughness, as shown below.

Figure 2 shows the comparison between the measured NR of a single DMPC bilayer and reflectivity calculated directly from an MD simulation. Very good agreement can be observed since the MD curves match all four measured contrasts simultaneously. As the simulations were run without a solid support and the silicon was added by hand while building the SLD profile, the water layer between the substrate and phospholipid head groups also has to be adjusted to fit the experimental data [36]. The layer being about 1 nm thick is in agreement with the literature [74]. The effect of changing this thickness on NR is also shown in Figure 2 (dashed lines). The 5 Å thinner water layer considerably flattens the bump in the reflectivity. As shown below, the influence of this water layer on NR is comparatively minor for multilamellar phospholipid systems.

There are several parameters related to the experimental setup that have to be taken into consideration when calculating reflectivities from an MD simulation, such as the instrumental resolution, background scattering and substrate roughness. It can also be seen that only two contrasts (D${}_{2}$O and 4 MW) exhibit a critical edge and that only the D${}_{2}$O measurement covers it with data points. This means that, for all but the D${}_{2}$O measurement, one has to rely on the scaling of the measured intensities to absolute values.

### 4.2. Neutron Reflectivity of a SoyPC Multilayer

The experimented presented in Section 4.1 demonstrated the methodology used on a single phospholipid bilayer. When simulating membrane fusion or stalk formation, of course at least two membranes are required, but, given the low density of stalks, experimentally multilayers (some tens to thousands of bilayers) have to be measured in order to obtain a detectable signal. MD simulations of this many bilayers are neither practically feasible nor useful, since the multilayer can be constructed by repeating the SLD profile of a single bilayer a suitable number of times. This section focuses on the main new features observed and the data evaluation challenges encountered during the study of a multilayer via reflectometry. As an exemplary system, multilamellar SoyPC was chosen since it was hypothesized that, for this phospholipid mixture, the presence of a drug promotes stalks formation [31], and, before MD and scattering methods can be used to look into the details of this question, a good description of the pure and unperturbed system is needed. SoyPC is a mixture of five major lipid components; only the most abundant polyunsaturated 1,2-dilinoleoyl-sn-glycero-3-phosphocholine (which we indicate in the following with DLPC, to not be confused with saturated 1,2-dilauroyl-sn-glycero-3-phosphocholine) was simulated by MD.

As in the previous case of DMPC, two DLPC bilayers separated by a water layer were simulated and the obtained SLD is presented in Figure 3 for two different contrasts. Once the SLD profile of a single bilayer is extracted from the MD, a multilayer is straightforwardly built by simply repeating the SLD profile *n* times in *z* direction. In the case of DLPC, 36 phospholipid bilayer repetitions were used. When applying such a perfect periodicity, this manual merging of two repetitions (usually in the water region) must be performed with caution so that the thickness of each water layer stays the same. Otherwise, an artificial rupture of symmetry can be introduced and the lattice constant would then be doubled. This would introduce a new peak in the calculated NR, at a *Q* position corresponding to half that of the first Bragg (demonstrated in Figure 4 as a dashed red peak). It is, however, also easily possible to introduce a certain degree of disorder in this step by adding randomness to the water layer thicknesses. In order to simulate the reflectivity in this case, one has to produce a large number of such structures and average the simulations. Such an incoherent addition is valid here since it is expected that the lamellar fluctuations and the corresponding interlamellar distances should be uncorrelated [75].

The SLD profile obtained for multilamellar DLPC in D${}_{2}$O was then used to calculate the reflectometry curve. The comparison with the experimental data is given in Figure 4. It is obvious that the simulated reflectivity (blue line) reproduces the first Bragg peak and fits the data well up to ≈0.1 Å${}^{-1}$. Between 0.1 and 0.2 Å${}^{-1}$, a discrepancy can be observed.

The measurements of SoyPC were in the past evaluated with an analytical model consisting of water, a head group region and a tail group region [31]. The different regions were represented by Gaussian density distributions and repeated without disorder. The algorithm then varied the flexible parameters of the model, e.g., layer thicknesses, etc. The data could be fitted well (green line in Figure 4) and the parameter results were very reproducible and independent of the starting parameters: The phospholipid tail region was fitted to be only 11 Å thick—a surprisingly small value. The water layer was 42 Å thick. The size of the unit cell (the repeat distance of the bilayers) is very well constrained (66 Å) by the positions of the Bragg peaks in the data. The unit cell size was therefore kept constant by the fit, proportionally enlarging the water layer between bilayers when reducing the bilayer thickness.

By comparing the MD SLD profile in Figure 3 with the analytical model published in [31], it is apparent that the analytic fit of the data proposes a smaller membrane thickness than the MD simulation. The water layer thickness in the simulation must be defined a priori by the number of water molecules between the adjacent lipid bilayers in the simulation box. However, this part of the SLD profile can (and must) be adjusted during the SLD profile modeling. Taking the MD model as a starting structure and adjusting the layer thicknesses, it was possible to reproduce the experimental data (see inset in Figure 4). This suggests that the force field and simulation parameters have to be adjusted to increase the density and reduce the thickness of the hydrophobic region. It might also mean that the MD simulation of a pure DLPC system is structurally still different from the mixture present in SoyPC.

Another feature worth noting is shown in Figure 5, which includes a zoom on the critical edge region. The exact position of the critical edge for total reflection is a function of the SLD difference between the semi-infinite medium on which the beam is reflected and the semi-infinite surroundings from which the beam comes. In our case, the beam comes through the side of a thick silicon wafer and is reflected at the interface with D${}_{2}$O. The exact shape of the reflectivity decay is obviously influenced by the additional layers, but the value of $Q_{c}$ below which the beam is totally reflected must remain the one of the material combination Si–D${}_{2}$O. A deviation of the critical edge from the theoretical position leads to the suspicion of a possible contamination of the D${}_{2}$O by hydrogenated molecules. Two possibilities of hydrogenated molecules come to mind: normal water (H${}_{2}$O) or phospholipids detached from the multilayer whose hydrogenated tails would significantly lower the overall SLD. In the case of H${}_{2}$O contamination, one would expect it to be homogeneously distributed across all hydrated parts of the sample. In the case of the phospholipid contamination, however, the contamination would be confined to the bulk water. Figure 5 shows the effect of both scenarios on the NR starting from the MD simulated SLD.

The blue reflectivity curve is obtained by scaling the bulk SLD of D${}_{2}$O by a factor of 0.8 on account of detached phospholipids with hydrogen-rich tails diffusing to the bulk. In this case, the SLD of D${}_{2}$O between the bilayers was not scaled. The red dashed curve is obtained by scaling the SLD of all D${}_{2}$O molecules by a factor of 0.8, simulating D${}_{2}$O contamination with H${}_{2}$O during the experiments. Since these corrections did not affect the interlamellar distance, the position of the Bragg peaks is not affected by this change in contrast. Scaling down the D${}_{2}$O SLD moves $Q_{c}$ close to the observed value. Adjusting the water SLD inside the lamellar structure, as in the hypothesis of light water contamination, has the effect of reducing the overall contrast of the lamellae and affects the Bragg peak intensity and reflectivity in the 0.1–0.15 Å${}^{-1}$ region. One could hope to be able to discriminate two solvent contamination origins on this basis. However, as shown by the green dashed curve in Figure 5, tuning the water SLD has a similar effect on the reflectivity as reducing the number of bilayers in the multilayer.

Neutron reflectivity measurements and reflectivity calculated directly from the MD simulations hint at the different structure of the investigated SoyPC multilayer. While the phospholipid bilayer thickness obtained from MD is larger than allowed by the position of the experimental Bragg peaks, the direct unconstrained fitting of the reflectivity using the analytical model suggests an extremely thin hydrophobic region (11 Å). From the MD point of view, such high compression seems hardly achievable since the hydrophobic region in that case has to be thinner than the polar heads. Additional experimental data at different contrasts could help to lift the ambiguity. Since these experimental data were not available, small-angle neutron scattering measurements of a single SoyPC bilayer were compared with the MD simulations.

### 4.3. Small-Angle Neutron Scattering of SoyPC Bilayers

Scattering cross sections obtained from SANS experiments on SoyPC in heavy water are reported in Figure 6. The data can be fit very well with a model of spherical unilamellar vesicles with polydisperse solvent cores [76,77], which is also expected based on the preparation method. A very careful inspection of the data revealed the presence of a small contribution of multilamellar vesicles [31], which can be neglected in the *Q* range presented here.

The best fit of the analytical model yields vesicles with a double-layer thickness of (33.0 ± 1.2) Å. The SLD profile obtained from the MD simulations was used for a comparison to the data. It was inserted into a model of *n* concentric spherical shells; the inner radius of the vesicle and their polydispersity was optimized by a fitting procedure. The results are displayed with a continuous red curve in Figure 6. There is a clear discrepancy between the experimental data and the description provided by the MD results, which is mainly due to a mismatch of the total membrane thickness. In particular, the oscillation at Q≈0.25 Å${}^{-1}$ is quite sensitive to this parameter. This is illustrated in the inset of Figure 6, where two dashed curves represent adding and subtracting 2.0 Å to the optimized bilayer thickness: a small change shifts the oscillation to higher or lower *Q*-values. The shoulder at $Q\approx 5\times 10^{-3}$ Å${}^{-1}$ is in contrast not sensitive to the change in bilayer thickness at all and is determined by the total size of the vesicles.

The SANS data on unilamellar vesicles clearly favor a thinner membrane than what is simulated by MD. The associated tail thickness of only 11 Å is, however, so incredibly thin that additional measurements on the pure DLPC system and with a variety of contrasts should be performed before addressing an optimization of the MD force field.

## 5. Discussion

Both reflectometry and small-angle scattering are low-resolution techniques. Their spatial sensitivity is limited to about $d_{\min}∼2\pi /Q_{\max}\approx 2\pi /\left(0.3\;{Å}^{-1}\right)\approx 20\;Å$. Smaller features can still change the scattering pattern if they change the average SLD of the layer in which they are embedded, but the information content in the data will not confine the shape of this feature. Reflectometry has the advantage over small-angle scattering that the sample is aligned and it is therefore possible to probe the SLD profile along the membrane normal. The scattering signal in small-angle scattering is generally an orientational average.

Despite the limitations of the scattering data, they are some of the very few experimental windows into this nano-world, and it is easy to compare simulations to the data. This combination is particularly powerful since the simulations provide a model that is already heavily constrained by many external inputs via the force field, while the scattering data provide a sensitive indicator of the plausibility of the simulated structures. The ease of comparing simulated and measured scattering curves to each other can, however, lead to an inflated degree of trust—from an experimenter’s point of view in the simulations and from a simulator’s point of view in the measurements. In the following, we therefore raise the awareness of each of the two communities for the potential problems of the other one—while, and this cannot be stressed enough, unreservedly recommending this combination.

### 5.1. Reflectometry

For the comparison of the reflectivity curves calculated from analytical or numerical SLD profiles to measured data, one has to take into account instrumental and sample non-idealities.

Effects caused by the instrument vary between different instrument types (e.g., monochromatic or time-of-flight) and even between different instruments of the same type. A non-exhaustive list is given in the following.
Every instrument will have sources of background which contaminate the intensity with a more or less random noise. These can be independent of the experiment (e.g., the perfectly random detection of cosmic particles) or instrument setting related (e.g., scattering of the probing particles on air or windows in the beam—also random and scattering on slits—a usually more or less strongly peaked effect).When the sample is illuminated under a very shallow angle, it might happen that only a fraction of the beam actually illuminates the sample. This geometric effect will be a function of the incident angle and the real intensity distribution in the beam (usually treated as Gaussian). In some configurations, this function could be strongly wavelength dependent, due to the ballistic effect: on their way to the sample, long wavelength neutrons, being slower, fall more than the short wavelength ones under the action of gravitation. They will thus impinge on the sample at a different spot and slightly different incident angle (an effect which also has to be taken into account to properly evaluate *Q*).Both aforementioned effects contribute to a normalization issue: since *R* is a relative measurement, one must ascertain that the full incident intensity is accurately measured. This can cause practical problems since the primary beam intensity is always orders of magnitude more intense than the reflected one. Gross errors in this step can be detected if enough data points have been taken in the regime of total reflection, but more subtle effects such as the above-mentioned over-illumination are much more difficult to detect if they affect the region where reflectivity intrinsically varies. It should be stressed that it is quite easy to overlook significant systematic errors since the *R* value is usually plotted on a logarithmic scale with 5–6 orders of magnitude.The measured intensity can be described by the convolution of the ideal signal with the instrumental resolution function. This convolution smears the measured curve and limits the possibility to resolve adjacent features (oscillations, peaks) in *Q* space. In real space, this translates to an upper limit for the measurable layer thickness and sensitivity to long-range correlations. Typically, the instrumental resolution of neutron reflectometers ranges about 1–10% ΔQ/Q and consists of contributions of the often dominant wavelength uncertainty and the beam divergence.A very careful treatment of error propagation during data reduction of the counted intensities is needed in order to preserve the possibility to evaluate the statistical agreement between a simulation and experimental data. Obviously, the error bar validity issue is paramount when dealing with fitting methods, and this is even more so when the fitted data vary over several orders of magnitude, as is the case for both reflectivity and SANS [78].

The sample itself also contributes features to the scattering data that are not reproduced by the computation of the reflectivity from the SLD profile:The sample membrane in a reflectometry measurement has to be supported by a substrate, either solid or liquid. This substrate can have an influence on the membrane properties, such as its rigidity. Studies looking at embedding larger proteins into the membrane might even experience collisions between the proteins and the substrate [46]. Further, the surface of the substrate can be ill-defined. While a reasonably thin silicon oxide layer usually does not influence the scattering data too much, the surface roughness of the substrate has an immediate effect on the data and can render the data useless if the roughness is not controlled to be below ∼5 Å. Besides influencing the quality of the data, it is clear that large surface irregularities will also affect the membrane morphology. The other half-infinite side also adds possibilities for imperfections that might not be mapped into the simulation: the solvent (especially when deuterated) might be contaminated with another isotope—either from the experimental setup of channels leading to the sample chamber or by hydrogen or hydrogen-containing groups escaping from the sample layer. This might have happened in the DLPC multilayer presented here where an amount of bilayers could have detached from the multilayer and float in some form through the solvent, lowering its SLD.A more subtle point concerns the water contrast variation. In order to make use of the different contrasts that can be achieved by isotopic substitution, one has to assume that the exchange between hydrogen and deuterium changes only the scattering lengths and not the actual structure. This is generally a justifiable approximation. The density [79,80] and many of the molecular interactions do change between H${}_{2}$O and D${}_{2}$O [80,81], but mostly very slightly. These—usually small—changes that happen in the real sample will of course not be reproduced by only one simulation where the isotopic exchange is performed a posteriori by assigning different scattering lengths to the atoms.The sample layer itself can also deviate from the modeled version in several aspects: concerning the SLD, it is basically impossible for an experimenter to ascertain the deuteration degree of the purchased phospholipids. Further, the deposition of phospholipids on the substrate might not have produced the structure that was intended (e.g., a single bilayer)—either on the complete sample or as inhomogeneities within the membrane plane. Neutron reflectometry measurements probe a surface on the order of 10–100 cm${}^{2}$: it is rather unrealistic that a phospholipid layer would coat such a large area homogeneously. Last, inhomogeneities can of course also occur in the direction of the membrane normal, such as a disorder of the water layer thickness between neighboring membranes in a multilayer.The sample will not only scatter neutrons/X-rays into the specular spot, but will also itself contribute an isotropic background which will add up to the extrinsic background sources discussed above. In the case of neutron scattering, this sample-related background level is dominated by the incoherent scattering from hydrogen atoms in the sample and will therefore vary between different contrasts of a given system. In the case of X-rays, the diffuse background is generated by the inelastic Compton scattering. It is measured and subtracted from the signal together with the off-specular scattering (see below). A remaining *Q*-independent background has to be accounted for in the modeling.Most importantly, in reflectometry, the sample will also generate *off-specular* scattering. This scattering intensity is caused by fluctuations of the SLD profile parallel to the membrane due, for instance, to membrane fluctuations. In the current context, this additional intensity overlaps the purely specular signal and needs to be subtracted from the experimental values before *R* can be evaluated. The length scale of the fluctuations responsible for off-specular scattering is up to the micrometer regime [20], which renders, as hinted above, an evaluation from the computer simulations impossible. The usual approach is therefore to measure the scattered intensity on both sides of the specular condition near to it in order to then interpolate the background intensity. On modern instruments using bidimensional detectors, one does not need to perform any additional experiment since a whole range of reflected angles is being covered around the specular direction. Figure 7 shows the intensity distribution as a function of incident angle ($\theta_{\mathrm{incident}}$) and reflection angle ($\theta_{\mathrm{reflected}}$). The specular line is seen along the main diagonal ($\theta_{\mathrm{incident}}=\theta_{\mathrm{reflected}}$), and it shows the total reflection region at the smallest angles. Along this line, the intensity maxima correspond to the Bragg peaks. The most prominent feature of this intensity map is, however, the broad off-specular band which follows the condition $\theta_{\mathrm{incident}}+\theta_{\mathrm{reflected}}=\theta_{\mathrm{Bragg}}$, which in *Q* space translates to $Q_{z}=Q_{\mathrm{Bragg}}$. As hinted above, this intensity band is thus characteristic of the in-plane correlations of the structures responsible for the Bragg peak. The broad width of the Bragg peak in the reciprocal space shows that the bilayers are only coherent over very small length scales. Subtraction of the underlying off-specular signal is clearly a challenging task, especially in regions where the overall reflectivity is low, leading to poor statistics. One needs to be aware of the risk of introducing systematic deviations from the true specular reflectivity during this data reduction step.

### 5.2. Small-Angle Scattering

While there is no substrate in small-angle scattering experiments, the other issues mentioned for reflectometry also exist for this technique. For example, unilamellar vesicles might be neither as spherical nor as homogeneously unilamellar as assumed. If they were produced by extensive sonication, the phospholipid molecules might even have been damaged and lyso-phospholipids can be present in the sample [82].

Factors that do not play a role for the samples used in reflectometry but have to be considered in SAS include the vesicle size polydispersity, which is basically always modeled with rather simple assumptions, such as a the Schulz–Zimm distribution. It is absolutely possible that the real size distribution is far more complex and does not smear the features in the scattering curve in exactly the way that is modeled. In addition, the positioning of different vesicles with respect to each other has an—often subtle—influence on the scattering data. If the interaction between the vesicles is known, it can (and should) be taken into consideration in the model.

The effects of an isotropic background and instrumental resolution are very similar to those observed in reflectometry. The background due to the sample itself is usually accounted for in the modeling, while all other background contributions are subtracted from the data using measurements of the empty sample container and the intrinsic noise of the detector. The instrumental resolution has to be taken into consideration much in the same way as for reflectometry.

Instrumental challenges that are more specific to small-angle scattering than reflectometry are related to the calibration of the detector efficiency [83], especially when attempting to obtain absolute units for the data in order to measure concentrations. Additionally, a full small-angle scattering curve is often measured in several steps with varying collimation lengths and the individual curves have to be stitched together before modeling, mostly manually.

As discussed in the Introduction, the theoretical treatment of SANS curves relies on the applicability of the first Born approximation. Care must be taken to avoid multiple scattering as this is not incorporated in the theoretical evaluation of the data—unfortunately, the absence or presence of multiple scattering is usually not apparent in the data. A practical rule of thumb is to lower the concentration until the sample transmits at least roughly 85% of the beam without interaction.

### 5.3. Molecular Dynamics Simulations

As in the case of the experiments, there are several aspects of computer simulations that might not be immediately clear to the non specialist, which might lead to misinterpretation of the results. In contrast to the DMPC phospholipid, containing saturated fatty acid chains, where simulations match experimental data very well, there is a disagreement in the case of polyunsaturated DLPC (the main component of SoyPC). The DLPC lipid bilayer thickness and area per lipid obtained in the MD simulation are not in agreement with the NR and SANS measurements. The experiments suggest a larger area per lipid (75 Å${}^{2}$) than the MD simulations (60 Å${}^{2}$) using the GROMOS 54A7 force field. The same problem was observed for polyunsaturated 1-stearoyl-2-docosahexaenoyl-sn-glycerco-3-phosphocholine (SDPC) bilayers [84] where high-level quantum mechanical calculations are used to improve the force fields’ (CHARMM36) dihedral potential of neighboring double bonds. An approach was proposed by Marquardt et al. [8] constraining the average area per lipid while allowing the *z* axis to expand and contract. This issue, including the force field reparameterization, is out of the scope of this article and will be addressed in our future work on SoyPC, including the measurements and comparison of pure mono- and multilamellar DLPC with the simulations.

It is important to say that simulated SLD profiles cannot be used “as received” and several adjustments have to be made. A first crucial point is related to the thickness of water layers: since the number of water molecules is fixed at the start of the simulation, the water layer thickness will also be artificially defined by the MD simulation. However, this issue can easily be solved and the water layer thickness can be adjusted as a parameter during the SLD profile calculation while keeping all the other parameters untouched. There are other practical reasons why the simulated system does not perfectly describe the actual sample used during the neutron/X-ray experiments. The MD simulation will not include the substrate and even less its interface imperfections. Due to limited computation power, the simulated volume usually represents a small fraction of the actual sample and cannot reproduce large-scale features such as the radius of curvature or fluctuations. The same argument justifies why the simplest subelements of quasi-periodic systems are simulated and then artificially reproduced, as for instance a bilayer is simulated in details rather than a true multilayer system.

There are some more obvious reasons a simulation might not be a true representation of reality: in a complex lipid mixture, the composition of the system under study has to be drastically simplified. In the current example of SoyPC, it was approximated by the most abundant phospholipid. In addition, the effects of pH are hard to reproduce: while the pH clearly plays a major role in reality, a simulation box will contain only very few OH${}^{-}$/H${}_{3}$O${}^{+}$ ions if they are included at all—and the most common force fields will not allow the phospholipids to change their protonation state. Besides these factors that limit the realism of the simulations, two more craftsmanship-related issues have to be considered since MD is nothing more than a means of sampling the phase space. First, the simulation has to be equilibrated for a long enough time. Second, the production run has to cover a long enough time for the system to sample the configurations in the vicinity of the energy minimum with accurate statistics. Transitions over energy barriers or phase transitions can pose significant obstacles against sufficient sampling. Further, there are severe limitations in simulations when it comes to temperature and pressure: both of these can typically not be expected to have a 1:1 relation to the real physical quantities. Just to name one example, the freezing point of commonly used water models varies between 213 and 271 K [85]. The last, even more general comment is that all available force fields represent only a part of the interactions that happen in reality. Moreover, even for the interactions taken into account, the force fields are always not more than mere simplifying parameterization. One might be tempted to discard classical MD and make use of a priori more realistic methods based on first principles. Apart from the fact that computing power severely limits the applicability of those methods to a smaller number of atoms, it should be noted that even the ab initio molecular dynamics simulations, which explicitly deal with many more effects than the classical simulations, still contain a number of adjustable parameters that have to be chosen by the simulation operator.

One aspect which has only been discussed briefly here and deserves some more comments is related to lateral correlations in the plane of the sample (such as lamellar fluctuations, undulations and fusion sites, e.g., stalks). We mention this point in the discussion of data reduction and specifically of background subtraction. Obviously, this scattering transports very important information about the exact nature of the in-plane structure and treating it as mere background is wasting information. However, given the very large length scales involved, atomic MD cannot produce relevant data and other methods, such as coarse-grained simulations, need to be used. Once a model of the structure has been constructed, however, well-documented software packages already exist which could be used to compute the corresponding off-specular scattering patterns from the reconstructed SLD distribution [86].

## 6. Conclusions

In this study, we showed, on the basis of a set of actual examples, how the structures obtained from MD simulations can be used to compute the corresponding scattering patterns in SANS and NR. It appeared along the way that several oversimplifications and assumptions have to be carefully dealt with, notably in producing a reliable description of the sample involving some “details” which are not simulated by MD (the substrate in reflectometry, the multilayer, D${}_{2}$O/H${}_{2}$O contamination, substrate roughness, etc.).

Clearly, potential imperfections and intrinsic limitations of all the techniques have to be kept in mind and overconfidence in a single observation to draw conclusions is at best risky. In our experience, however, confronting the experiment to the simulation and *vice versa* is a beneficial process for both sides as it opens opportunities for further understanding of the systems under study and, on a more mundane level, it helps to detect and/or understand inconsistencies (such as solvent contamination, the importance of fluctuations, etc.).

The complementarity of scattering methods and MD simulations is striking, not only because it bridges the divide between direct and reciprocal space and helps to solve the age old phase problem, but also because each method sheds light in the blind spot of the other, for instance in terms of accessible length scales and timescales.

Apart from suggesting new experiments to be performed on this very system (e.g., monolayers and increasingly thick multilayer systems, more contrasts and eventually moving to more complex/interesting systems such as those including drugs), this work hints at possible methodological developments such as the systematic use of MD models for the preparation and analysis of scattering experiments.

The calculation of expected, reasonable scattering patterns can assist in the preparation and the optimization of experiments to be performed at large-scale facilities. Subtle instrumental effects could be simulated in the framework of virtual experiments such as those performed using Monte-Carlo simulation packages [87]. Knowing where to expect important features or how these would differ between competing models would allow measurements to be tailored to concentrate on these regions.

One can further imagine a range of tools that would allow the investigator to alter the MD simulation to optimize the agreement between calculated scattering curves and experimental data: first, one could tweak the SLD profile without touching the simulation itself, simply using it as a suitable starting point. Second, the deviation between calculated and measured scattering curves can be employed as an additional contribution to the potential, driving the simulation into a compatible configuration [88]. Third, it might be possible to adjust individual parameters in the force field, possibly via big data/machine learning approaches. Scattering methods are possibly the only class of experiments that probe directly the very thing MD simulates, giving a unique angle in this ambitious endeavor.

To facilitate these ideas, a new software for calculating neutron and X-ray small-angle scattering and reflectivity patterns directly from the MD simulation trajectory, *Made2Reflect*, will be published soon. This standalone Python program allows the fast and simple analysis of large trajectories and is applicable not only for phospholipid membranes but also for electrochemistry, corrosion and batteries, i.e., solid–liquid and liquid–liquid interfaces in general.

## Figures and Tables

**Figure 1 membranes-11-00507-f001:**
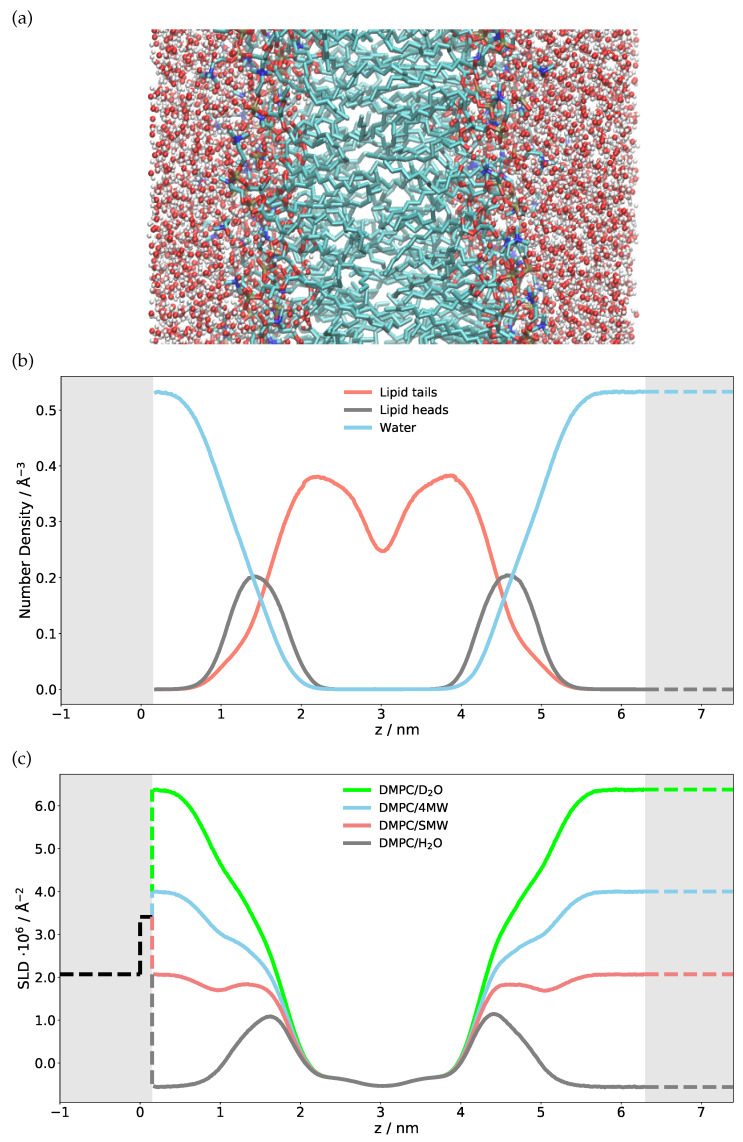
(**a**) Snapshot of the MD simulation of a single free-floating DMPC bilayer in water. The lipid tails can be seen in green, whereas the water molecules are red/white. (**b**) The extracted number density of atoms after averaging over the whole simulation time and summed together based on their presence in a certain group (water/heads/tails). (**c**) Neutron scattering length density (SLD) at four different contrasts calculated from the different atomic number densities. Shaded regions are hand-modeled SLD values for Si/SiO${}_{2}$ (left) and bulk solvent (right). The SLD can be negative.

**Figure 2 membranes-11-00507-f002:**
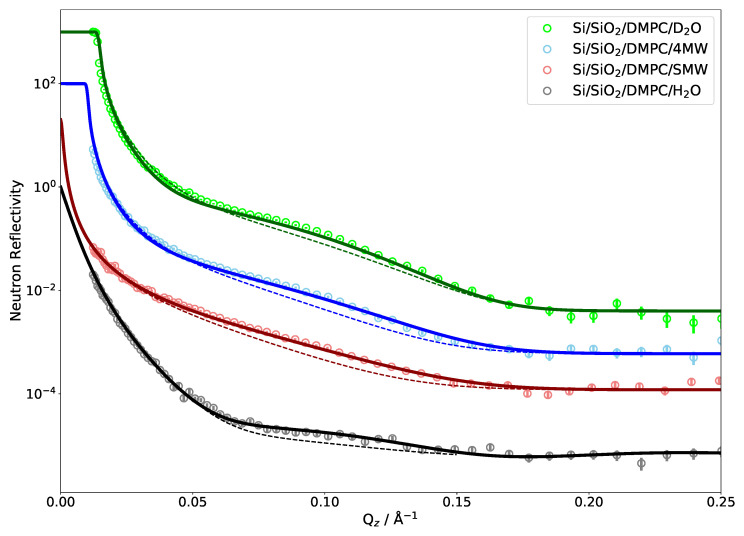
Neutron reflectivity of a DMPC bilayer on a silicon substrate in water for four different H/D contrasts. Points represent experimental data and lines are neutron reflectivity calculated directly from the MD simulations. The curves are shifted along the *y* axis by a factor 10 each for clarity; they all extrapolate to R(Q=0)=1. Dashed lines show the effect of reducing the water layer thickness between the substrate and phospholipid head groups from 10 to 5 Å.

**Figure 3 membranes-11-00507-f003:**
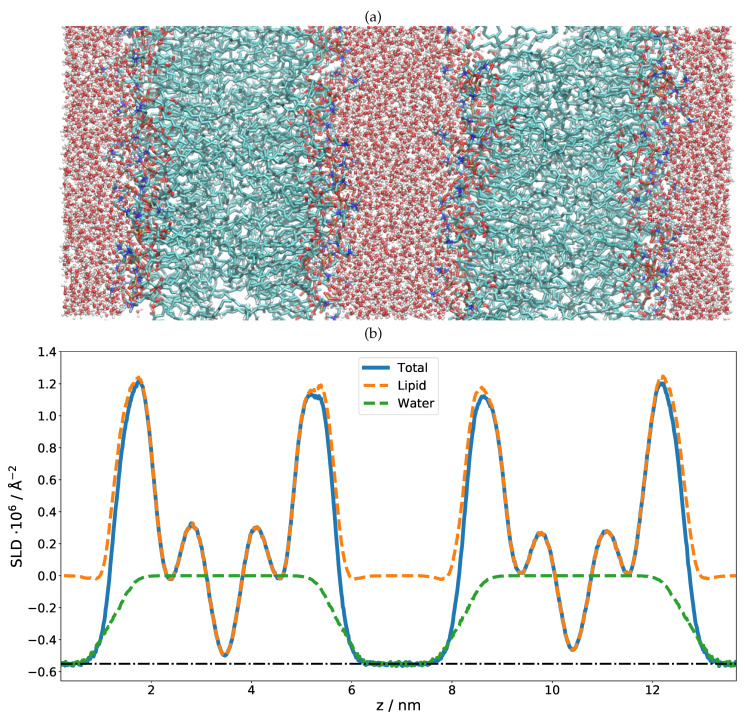
(**a**) MD simulation snapshot of the double phospholipid bilayer. (**b**) SLD profile of the first contrast, H${}_{2}$O/bilayer/H${}_{2}$O/bilayer/H${}_{2}$O. Note that there are negative SLD values. The dash-dotted line indicates the SLD of pure H${}_{2}$O. (**c**) SLD profile of the second contrast, D${}_{2}$O/bilayer/D${}_{2}$O/bilayer/D${}_{2}$O. The dash-dotted line indicates the SLD of pure D${}_{2}$O.

**Figure 4 membranes-11-00507-f004:**
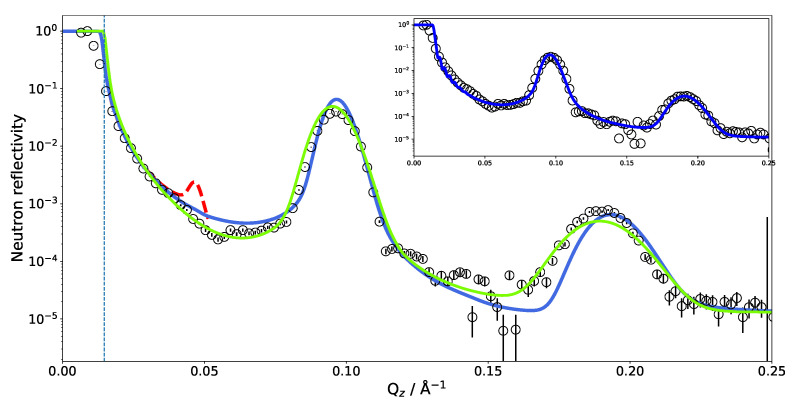
Experimental neutron reflectivity of SoyPC multilayer in D${}_{2}$O compared with different models. Green line, analytical model published by Mangiapia et al. [31]; blue line, neutron reflectivity calculated directly from the MD simulation. The dashed vertical line marks the theoretical $Q_{c}$ of Si/D${}_{2}$O. The dashed red peak reveals the artificial asymmetry in the repartition of the water layer thicknesses (details in the text). The blue line in the inset shows an adjusted MD model (details in the text).

**Figure 5 membranes-11-00507-f005:**
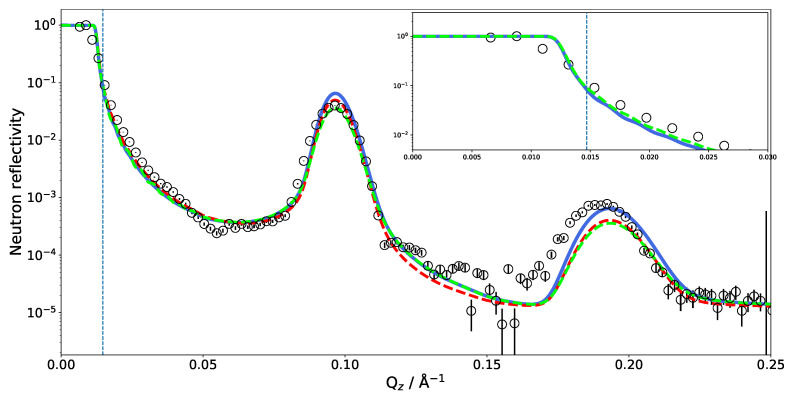
The influence of varying the SLD of D${}_{2}$O on the critical edge ($Q_{c}$) of NR calculated from the MD simulation. Blue line, SLD of only bulk D${}_{2}$O scaled by a factor of 0.8; dashed red line, SLD of all D${}_{2}$O in the system scaled by a factor of 0.8; dashed green line, SLD profile after reducing the number of layers in the modeled multilayer from 36 to 20. The dashed vertical line marks the theoretical $Q_{c}$ of Si/D${}_{2}$O.

**Figure 6 membranes-11-00507-f006:**
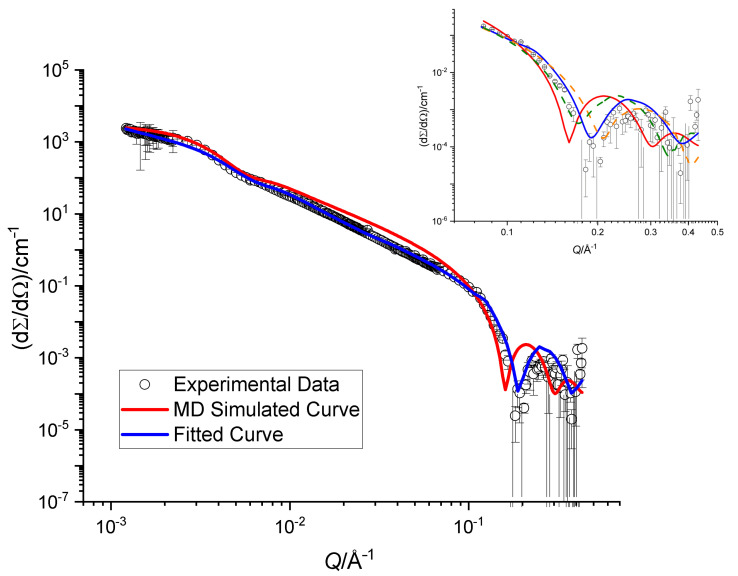
Scattering cross sections obtained for an extruded sample of SoyPC in D${}_{2}$O. The blue line corresponds to the theoretical cross sections obtained by fitting the model described in the text, whereas the red curve is obtained from the MD SLD profile. The inset shows a zoom on the high-*Q* region together with two additional dashed curves obtained by adding (in dark green) and subtracting (in orange) 2 Å to the value of the bilayer thickness extracted from the best fit.

**Figure 7 membranes-11-00507-f007:**
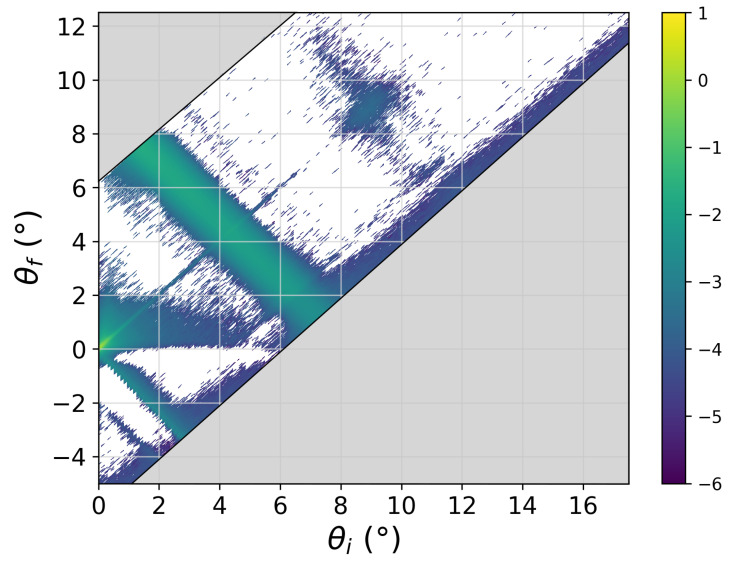
Off-specular reflectivity map (log scale) of the multilayer sample as a function of the angle of incidence ($\theta_{\mathrm{incident}}$) and of the reflection angle ($\theta_{\mathrm{reflected}}$). The specular reflected beam contributes only at $\theta_{\mathrm{incident}}=\theta_{\mathrm{reflected}}$. The width of the recorded band around the specular line is defined by the detector size.

## Data Availability

The data presented in this study are openly available: DMPC specular reflectivity (10.5286/ISIS.E.RB1820565 and 10.5281/zenodo.4882721). SoyPC specular and off-specular neutron reflectivity, SoyPC SANS data, DLPC force field parameters (Gromos54a7) and DLPC.pdb structure (10.5281/zenodo.4882721). Publicly available DMPC force field parameters were used in this study. This data can be found here: http://wcm.ucalgary.ca/tieleman/downloads, accessed on 1 June 2021.

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
