# Peer review of "Mutually Beneficial Combination of Molecular Dynamics Computer Simulations and Scattering Experiments"

_membranes, 2021, doi:10.3390/membranes11070507_

Round 1

Reviewer 1 Report

The article by Zec et al investigated PC-membrane systems using experiments and MD simulations. The arguments presented by the authors are convincing indeed in how either technique may experience shortcomings in probing a hypothesis. A complimentary approach in such as scenario certainly comes to the rescue in hopefully resolving contradictory viewpoints.

(1) The authors have chosen a DMPC system in this article. Amongst the many lipid bilayer systems or even PC-systems, why was this chosen specifically? It would be nice to have some more clarity on that.

(2) This may have been a typographical error, the units for area per lipid need to be corrected to A^2 in the line-497.

This work should certainly be read by a larger scientific community.

Author Response

The article by Zec et al investigated PC-membrane systems using experiments and MD simulations. The arguments presented by the authors are convincing indeed in how either technique may experience shortcomings in probing a hypothesis. A complementary approach in such as scenario certainly comes to the rescue in hopefully resolving contradictory viewpoints.

- DONE (1) The authors have chosen a DMPC system in this article. Amongst the many lipid bilayer systems or even PC-systems, why was this chosen specifically? It would be nice to have some more clarity on that.
****
There are 2 reasons why the DMPC lipid has been chosen. DMPC is a phospholipid widely used in many biological and biophysical investigations, as well as component in some pharmaceutical formulations. Being a well studied "simple" system, DMPC lends itself as an ideal candidate to validate the methodology proposed in the article. In the revised version of our manuscript, we have deepened this point. Secondly, DMPC is of particular interest for some of us since there is an ongoing study on DMPC - protein interactions.
****

- DONE (2) This may have been a typographical error, the units for area per lipid need to be corrected to A^2 in the line-497.
****
In the revised version of the manuscript, we have corrected the typographical errors. We thank the reviewer for the suggestions that benefited the manuscript
****

This work should certainly be read by a larger scientific community.

Reviewer 2 Report

This paper is basically a review article, showing the potential offered
by the combination of neutron and x-ray reflectometry and small angle
scattering with computer simulations.

While the paper does not contain any new scientific insight, as the results
shown have already been published previously and they are not further discussed here, they serve the purpose of serving as examples of the synergy between x-ray and neutron scattering and simulation. 

As a review paper, this manuscript is well written and presents a reasonably
complete overview of the experimental techniques (reflectometry and small
angle scattering) and the MD simulations, focusing in the main steps needed
to calculate reflectivity or SAS curves from the MD and the potential problems involved in the comparison between simulation and experiment. In particular, the final discussion about the experimental and simulation problems and the pitfalls that need to be taken into consideration when comparing both is a useful contribution to the field.

Therefore I consider that this manuscript can be published as it is.

There are only a few minor points where I do not agree with the authors and
I would like to ask them to reconsider the correctness of their affirmations.

They are:

In L. 62: "Neutron scattering and MD simulations essentially probe different time and length scales". This could be debated, and certainly depends on the particular neutron scattering technique and the available computing power. However, in general I would say that neutron scattering and MD time and length scales are quite similar, and they certainly overlap much more than for any other experimental technique.

How are eqs. (14)-(15) obtained? The standard Fresnel relations are of the form (n1*cos(th1)-n2*cos(th2))/(n1*cos(th1)+n2*cos(th2)), etc.
If eqs. (14)-(15) are correct, under which approximations are they valid?

In L. 839, I don't agree with the comment that instrument resolution is often neglectedin the evaluation of small angle scattering data. While this was  perhaps true in the past, nowadays most (if not all) of data reduction software outputs also the corresponding instrument resolution and good data analysis software will use this information in the fitting process. In the  same line, while it is true that many users keep trying to stitch together curves measured at different conditions, this is not an absolute necessity and
modern software allows loading several curves and fitting them simultaneously.

Finally, the results concerning the SoyPC multilayer remain quite puzzling and the manuscript gives more questions than answers, raising doubts on both the experimental and the simulation results. However, it seems that at present the authors are not in a position of giving a more precise view and taking into account the nature of the paper this can be accepted and taken as an example of the limits discussed in section 5.

Other suggested corrections are:

L. 30: plays --> play
L. 81: in AN effective
L. 110: as an VMD --> as a VMD
L. 224: From those observation --> observations
L. 295: lacks in the ability --> lacks the ability
L. 390: i.a.?
L. 474: 3000 SPC water molecules between the layers was simulated at NPT conditions?
L. 480-481: The time-step is repeated. And even a 3rd time in L. 490.
L. 512: obtaining A scattering length density profile
L. 594: produce a large number of such structureS
L. 658: the analytical model suggests AN extremely thin hydrophobic region
L. 741: limits the possibility to resolve of adjacent features --> remove of
L. 789: later --> layer
L. 859: are not in an agreement --> are not in agreement
L. 885: mixtures --> mixture

Author Response

This paper is basically a review article, showing the potential offered by the combination of neutron and x-ray reflectometry and small angle scattering with computer simulations. While the paper does not contain any new scientific insight, as the results shown have already been published previously and they are not further discussed here, they serve the purpose of serving as examples of the synergy between x-ray and neutron scattering and simulation. As a review paper, this manuscript is well written and presents a reasonably complete overview of the experimental techniques (reflectometry and small angle scattering) and the MD simulations, focusing in the main steps needed to calculate reflectivity or SAS curves from the MD and the potential problems involved in the comparison between simulation and experiment. In particular, the final discussion about the experimental and simulation problems and the pitfalls that need to be taken into consideration when comparing both is a useful contribution to the field. Therefore I consider that this manuscript can be published as it is. There are only a few minor points where I do not agree with the authors and I would like to ask them to reconsider the correctness of their affirmations.
They are:

In L. 62: "Neutron scattering and MD simulations essentially probe different time and length scales". This could be debated, and certainly depends on the particular neutron scattering technique and the available computing power. However, in general I would say that neutron scattering and MD time and length scales are quite similar, and they certainly overlap much more than for any other experimental technique.
****
We agree with the reviewer and have revised the misleading section. Indeed, if the probed time and length scales were completely distinct from each other, a comparison of these techniques would be impossible. We hope to have made clear in the revised version that there is an overlap in the covered time and length scales, and in addition each of the techniques can reach times and lengths that are inaccessible for the other.
****

How are eqs. (14)-(15) obtained? The standard Fresnel relations are of the form (n1*cos(th1)-n2*cos(th2))/(n1*cos(th1)+n2*cos(th2)), etc.
If eqs. (14)-(15) are correct, under which approximations are they valid?
****
The Fresnel equations are derived from the continuity conditions as follows:
continuity equation for the wave at the interface:
aI+aR = aT
and vertical component of the continuity equation for the derivative of the wave
-(aI-aR)k sinα= -aT(nk) sinα'
lead to
(aI-aR)/(aI+aR) = n  sinα/sinα'  

Under the small angle approximation (valid in reflectometry) and given the fact that n is almost 1 for x-rays and neutrons, one gets equations 14 and 15 which provide a compact and easy to grasp formulation for the amplitude reflectivities and transmitivities which removes the here irrelevant trigonometric accuracy of the more general forms involving sin or tan.
Reference 44 provides the full development.

We acknowledge that the small angle approximation should be mentioned and have corrected the text accordingly.
****

In L. 839, I don't agree with the comment that instrument resolution is often neglected in the evaluation of small angle scattering data. While this was  perhaps true in the past, nowadays most (if not all) of data reduction software outputs also the corresponding instrument resolution and good data analysis software will use this information in the fitting process. In the  same line, while it is true that many users keep trying to stitch together curves measured at different conditions, this is not an absolute necessity and modern software allows loading several curves and fitting them simultaneously.
****
In the revised version of the manuscript we have addressed the question raised by the reviewer, modifying the related part and improving the involved sentences.
****

Finally, the results concerning the SoyPC multilayer remain quite puzzling and the manuscript gives more questions than answers, raising doubts on both the experimental and the simulation results. However, it seems that at present the authors are not in a position of giving a more precise view and taking into account the nature of the paper this can be accepted and taken as an example of the limits discussed in section 5.

We agree with the reviewer that a more consistent picture of the SoyPC structure would have been desirable. Nevertheless, we would like to stress again that it is only because of the comparison between simulation and experiment that these questions arise -- if only one of the techniques had been employed, one could have drawn a clear but probably wrong conclusion.

Other suggested corrections are:

- DONE L. 30: plays --> play
- DONE L. 81: in AN effective
- DONE L. 110: as an VMD --> as a VMD
- DONE L. 224: From those observation --> observations
- DONE L. 295: lacks in the ability --> lacks the ability
- DONE L. 390: i.a.? - changed to: inter alia
- DONE L. 474: 3000 SPC water molecules between the layers was simulated at NPT conditions? - Yes, cf. text.
- DONE L. 480-481: The time-step is repeated. And even a 3rd time in L. 490.
- DONE L. 512: obtaining A scattering length density profile
- DONE L. 594: produce a large number of such structureS
- DONE L. 658: the analytical model suggests AN extremely thin hydrophobic region
- DONE L. 741: limits the possibility to resolve of adjacent features --> remove of
- DONE L. 789: later --> layer
- DONE L. 859: are not in an agreement --> are not in agreement
- DONE L. 885: mixtures --> mixture

We would like to thank the reviewer for their thoughtful comments and hope that the changes we introduced in the manuscript address all the raised points.